# Bag of Instances Aggregation Boosts Self-supervised Distillation

**Haohang Xu**[1,2*] **Jiemin Fang**[3,4*] **Xiaopeng Zhang**[2] **Lingxi Xie**[2]
**Xinggang Wang**[4] **Wenrui Dai**[1] **Hongkai Xiong**[1] **Qi Tian**[2]
[1]Shanghai Jiao Tong University   [2]Huawei Inc.
[3]Institute of Artificial Intelligence, Huazhong University of Science & Technology
[4]School of EIC, Huazhong University of Science & Technology
{xuhaohang, daiwenrui,xionghongkai}@sjtu.edu.cn
{jaminfong, xgwang}@hust.edu.cn
{zxphistory, 198808xc}@gmail.com  tian.qi1@huawei.com

## Abstract

Recent advances in self-supervised learning have experienced remarkable progress, especially for contrastive learning based methods, which regard each image as well as its augmentations as an individual class and try to distinguish them from all other images. However, due to the large quantity of exemplars, this kind of pretext task intrinsically suffers from slow convergence and is hard for optimization. This is especially true for small scale models, which we find the performance drops dramatically comparing with its supervised counterpart. In this paper, we propose a simple but effective distillation strategy for unsupervised learning. The highlight is that the relationship among similar samples counts and can be seamlessly transferred to the student to boost the performance. Our method, termed as BINGO, which is short for **B**ag of **I**nsta**N**ces a**G**gregati**O**n, targets at transferring the relationship learned by the teacher to the student. Here bag of instances indicates a set of similar samples constructed by the teacher and are grouped within a bag, and the goal of distillation is to aggregate compact representations over the student with respect to instances in a bag. Notably, BINGO achieves new state-of-the-art performance on small scale models, *i.e.*, 65.5% and 68.9% top-1 accuracies with linear evaluation on ImageNet, using ResNet-18 and ResNet-34 as backbone, respectively, surpassing baselines (52.5% and 57.4% top-1 accuracies) by a significant margin. The code is available at https://github.com/haohang96/bingo.

## 1 Introduction

Convolutional Neural Networks (CNNs) have achieved great success in the field of computer vision, including image classification (He et al., 2016), object detection (Ren et al., 2015) and semantic segmentation (Chen et al., 2017). However, most of the time, CNNs cannot succeed without enormous human-annotated data. Recently, self-supervised learning, typified by contrastive learning (He et al., 2020; Chen et al., 2020a), has been fighting with the annotation-eager challenge and achieves great success. Most current self-supervised methods yet focus on networks with large size, *e.g.*, ResNet-50 (He et al., 2016) with more than 20M parameters, but real-life implementation usually involves computation-limited scenarios, *e.g.*, mobile/edge devices.

Due to annotation lacking in unsupervised tasks, learning from unlabeled data becomes challenging. Recent contrastive learning methods (He et al., 2020; Chen et al., 2020a) tackle this problem by narrowing gaps between embeddings of different augmentations from the same image. Techniques like momentum encoder for stable updating, memory bank for storing negative pairs, complicated data augmentation strategies *etc*., are proposed to avoid collapse and promote the performance. With the above techniques, contrastive learning methods show promising performance. However,

---

*Equal contributions.
The work was done during the internship of Haohang Xu and Jiemin Fang at Huawei Inc.

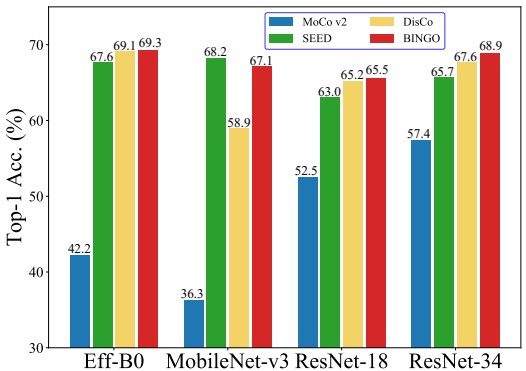 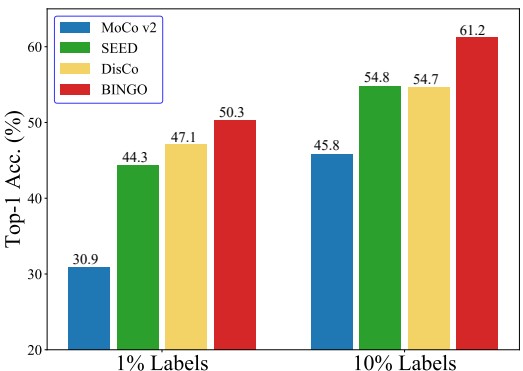

(a) Linear classification accuracy on ImageNet over different student architectures distilled by ResNet-50×2 teacher model

(b) Semi-supervised learning by fine-tuning 1% and 10% labeled images on ImageNet using ResNet-18 student and ResNet-152 teacher model

Figure 1: Overall performance comparisons between BINGO and other unsupervised distillation methods.

contrastive learning requires discriminating all instances, due to the large quantity of exemplars, this kind of pretext task intrinsically suffers from slow convergence and is hard for optimization. This issue becomes severe for small scale models, which carry too few parameters to fit the enormous data. Inspired by supervised learning that knowledge from large models can effectively promote the learning ability of small models with distillation, exploring knowledge distillation on unsupervised small models becomes an important topic.

Compress (Fang et al., 2020) and SEED (Fang et al., 2020) are two typical methods for unsupervised distillation, which propose to transfer knowledge from the teacher in terms of similarity distributions among different instances. However, as the similarity distribution is computed by randomly sampling instances from a dynamically maintained queue, this kind of knowledge is mostly constructed based on instances with low relation, which fails to effectively model similarity of those highly related samples. To solve this issue, we propose a new self-supervised distillation method, which transfers knowledge by aggregating bags of related instances, named **BINGO**. In our empirical studies, transferring knowledge based on highly related samples helps boost performance more effectively compared with previous relation-agnostic methods. Specifically, we select an unsupervised pretrained large model as the teacher. First, we map the conventional instance-wise dataset into a bag-wise one. Each original instance is set as an *anchor instance* of the bag. By matching similarities of all the other instances' embeddings produced by the teacher model, we feed instances which show high similarity with the anchor instance into the bag. Then we apply the bagged dataset to the small model distillation process. To this end, we propose a *bag-aggregation* distillation loss, which consists of two components: *inter-sample* distillation and *intra-sample* distillation. For intra-sample distillation, embeddings of the student and teacher from two augmentations of the same instance are pushed together; for inter-sample distillation, embeddings of all instances in one bag are pushed to be more similar with the anchor one. Equiped with the two proposed distillation loss, the bag-based knowledge from the teacher can be well transferred to the student, which shows significant advantages over previous relation-agnostic ones (Fang et al., 2020; Abbasi Koohpayegani et al., 2020).

Our contributions can be summarized as follows.

- We propose a new self-supervised distillation method, which bags related instances by matching similarities of instance embeddings produced by the teacher. The bagged dataset can effectively boost small model distillation by aggregating instance embeddings in bags. The proposed relation-guided method shows stronger performance than previous relation-agnostic ones.

- BINGO promotes the performance of both ResNet-18 and -34 to new state-of-the-art (SOTA) ones in unsupervised scenarios. It is worth noting that the distilled models also present far better performance compared with previous SOTA methods on other tasks, *i.e.*, KNN classification and semi-supervised learning.

- BINGO provides a new paradigm for unsupervised distillation where knowledge between instances with high relation could be more effective than relation-agnostic ones. This may be inspiring for further explorations on knowledge transfer in unsupervised scenarios.

## 2 RELATED WORK

**Self-supervised Learning**   As a generic framework to learn representations with unlabeled data, self-supervised learning has experienced remarkable progress over the past few years. By constructing a series of pretext tasks, self-supervised learning aims at extracting discriminative representations from input data. Previous methods obtain self-supervised representations mainly via a corrupting and recovering manner, from perspectives of spatial ordering (Noroozi & Favaro, 2016), rotation changes (Komodakis & Gidaris, 2018), in-painting (Pathak et al., 2016), or colorization (Zhang et al., 2016), *et al*. Recently, contrastive learning based methods (He et al., 2020; Chen et al., 2020a) emerge and significantly promote the performance of self-supervised learning, which aim at maximizing the mutual information between two augmented views of a image. A series of subsequent works (Grill et al., 2020; Xu et al., 2020b; Dwibedi et al., 2021) further improve the performance to a very high level. Khosla et al. (2020) applies contrastive learning on supervised learning, which selects the positive samples from the same category. Caron et al. (2020) proposes to align the distribution of one instance's different views on other categories. However, few of them pay attention to self-supervised learning on small-scale models, which are of critical importance to implement self-supervised models on lightweight devices. We propose an effective method to boost the self-supervised learning of small models, which takes advantage of relation-based knowledge between data and shows superior performance than previous ones.

**Knowledge Distillation**   Knowledge distillation aims to transfer knowledge from a model (teacher) to another one (student), usually from a large to small one, which is commonly used for improving the performance of the lightweight model. Hinton et al. (2015) first proposes knowledge distillation via minimizing the KL-divergence between the student and teacher's logits, which uses the predicted class probabilities from the teacher as soft labels to guide the student model. Instead of mimicking teacher's logits, Romero et al. (2014) transfers the knowledge by minimizing the $\ell_2$ distance between intermediate outputs of the teacher and student model. To solve the dimension mismatch, Romero et al. (2014) uses a randomly initialized projection layer to enlarge the dimension of a narrower student model. Based on Romero et al. (2014), Zagoruyko & Komodakis (2016) utilizes knowledge stored in the attention map generated by the teacher model, and pushes the student model to pay attention to the area where the teacher focuses on. Zhou et al. (2021) improves weighted soft labels to adaptively improve the bias-variance tradeoff of each sample. Besides perspectives of soft labels and intermediate features, relation between samples is also an important knowledge. Park et al. (2019) and Liu et al. (2019) train student model by aligning the pair-wise similarity graph with the teacher. Recently, some works extend the above distillation method into self-supervised learning scenarios. Tian et al. (2019) uses the contrastive loss to learn cross-modality consistency. Xu et al. (2020a),Fang et al. (2020) and Abbasi Koohpayegani et al. (2020) share a similar methodology with Caron et al. (2020) of aligning feature distribution between views of the same instances. The distribution is computed as the pair-wise similarities between student's outputs and features stored in memory bank. However, the above relation-based self-supervised distillation methods only compute the similarity between anchor sample and randomly sampled instances from a maintained queue, which ignores the relation between sampled and anchor instances. Choi et al. (2021) uses the teacher model to produce cluster assignments, and encourages the student model to mimic the output of the trainable teacher model on-the-fly, which achieves promising results. Gao et al. (2021) strengthens the student model by adding a regularization loss on the original contrastive loss, which aims at minimizing the $\ell_2$ distance between the student's and teacher's embedding. Navaneet et al. (2021) also achieves competitive results with feature regression in self-supervised distillation. We propose to transfer the relation knowledge between models via a new type of dataset, which bags related instances. By aggregating the bagged instances, the relation knowledge can be effectively transferred.

## 3 APPROACH

In this section, we introduce the proposed BINGO in details. First, we discuss how to bag samples in the instance-wise dataset. After the samples are bagged, the bag-aggregation based knowledge

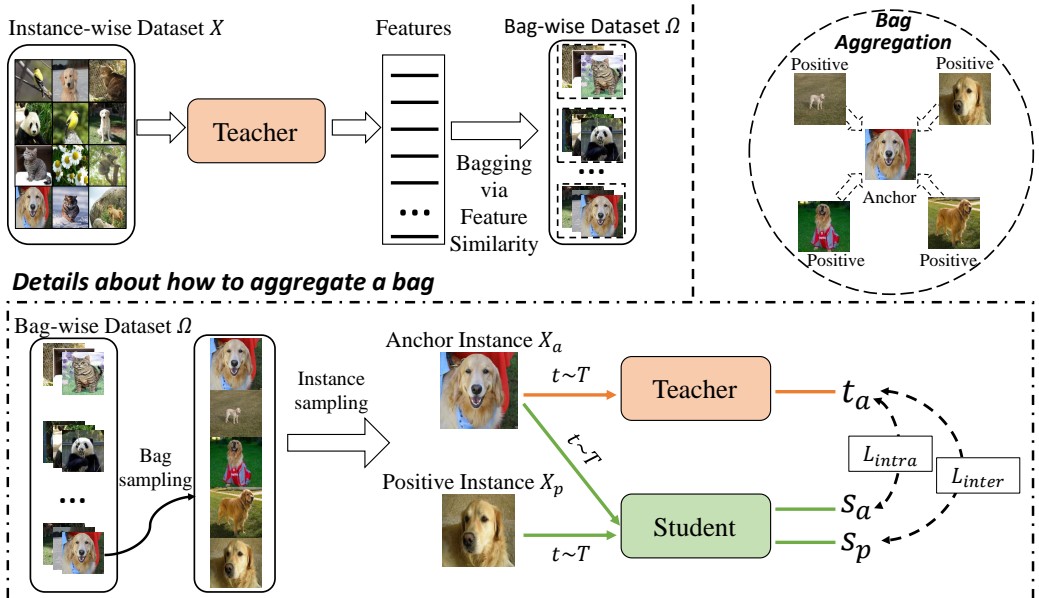

Figure 2: An overview of the proposed method. The samples are first bagged via feature similarity. Then the related instances in a bag is aggregated via intra-sample and inter-sample distillation loss. The figure on top-right is an intuitive explanation of how bag aggregation works.

distillation is introduced. We also discuss how to compute bag-aggregation loss and how they improve the performance of the lightweight model. The overall framework is illustrated in Fig. 2.

### 3.1 BAGGING INSTANCES WITH SIMILARITY MATCHING

Given the unlabeled training set $X = \{\mathbf{x_1}, \mathbf{x_2}, ..., \mathbf{x_N}\}$, we define the corresponding bag-wise training set as $\mathbf{\Omega} = \{\mathbf{\Omega_1}, \mathbf{\Omega_2}, ..., \mathbf{\Omega_N}\}$, where each bag $\mathbf{\Omega_i}$ consists of a set of instances. To transfer the instance-wise dataset to a bag-wise one, we first feed $X$ into a pretrained teacher model $f_\mathbf{T}$ and get the corresponding features $V = \{\mathbf{v_1}, \mathbf{v_2}, ..., \mathbf{v_N}\}$ where $\mathbf{v_i} = f_\mathbf{T}(\mathbf{x_i})$. For each anchor sample $\mathbf{x}_a$ in the dataset, we find positive samples which share high similarity with the anchor sample. Then the anchor sample as well as the similar samples are combined to form a bag. The samples in one bag have a compact representation in the embedding space. Several mapping function can be used to find similar samples:

**K-nearest Neighbors** For each anchor sample $\mathbf{x}_a$ in the instance-wise dataset, we first compute the pairwise similarity with all samples in the dataset $\mathbf{S}_a = \{\mathbf{v}_a \cdot \mathbf{v_i} \mid \mathbf{i} = 1, 2, ..., N\}$. The bag $\mathbf{\Omega}_a$ corresponding to $\mathbf{x}_a$ is defined as:

$$\mathbf{\Omega}_a = \mathbf{top{-}rank}(\mathbf{S}_a, \mathbf{K}), \tag{1}$$

where $\mathbf{top{-}rank}(\cdot, \mathbf{K})$ returns the indices of top $\mathbf{K}$ items in a set.

**K-means Clustering** Given the training feature set $V = \{\mathbf{v_1}, \mathbf{v_2}, ..., \mathbf{v_N}\}$, we first assign a pseudo-label $\mathbf{q_i}$ to each sample $\mathbf{i}$, where $\mathbf{q_i} \in \{\mathbf{q_1}, ..., \mathbf{q_K}\}$. The clustering process is performed by minimizing the following term,

$$\frac{1}{N} \sum_{i=1}^{N} -\mathbf{v_i^T} \mathbf{c_{q_i}}, \tag{2}$$

where $\mathbf{c_{q_i}}$ denotes the centering feature of all features belonging to the label $\mathbf{q_i}$, *i.e.*, $\mathbf{c_{q_i}} = \sum_{\mathbf{q_j} = \mathbf{q_i}} \mathbf{v_j}, \forall j = 1, ..., N$.

The bag $\mathbf{\Omega}_a$ of anchor sample $\mathbf{x}_a$ is defined as:

$$\mathbf{\Omega}_a = \{\mathbf{i} \mid \mathbf{q_i} = \mathbf{q}_a, \forall \mathbf{i} = 1, 2, ..., N\}. \tag{3}$$

**Ground Truth Label** If the ground truth label is available, we can also bag samples with the human-annotated semantic labels. Given the label set $Y = \{\mathbf{y_1}, \mathbf{y_2}, ..., \mathbf{y_N}\}$, we can bag related instances of the anchor sample $\mathbf{x}_a$ via:

$$\mathbf{\Omega}_a = \{\mathbf{i} \mid \mathbf{y_i} = \mathbf{y}_a, \forall \mathbf{i} = 1, 2, ..., N\}. \tag{4}$$

In this paper, we use K-nearest neighbors as the bagging strategy. More details about performance of using the K-means clustering based bagging strategy can be found in Appendix. Note that bagging instances via the ground truth label is just used to measure the upper bound of the proposed method.

### 3.2 KNOWLEDGE DISTILLATION VIA BAG AGGREGATION

Once we get the bag-wise dataset $\mathbf{\Omega}$ utilizing a pretrained teacher model, it can be used for distillation process. In each feed-forward process, the anchor sample $\mathbf{x}_a$ and the positive sample $\mathbf{x}_p$ which belong to the same bag $\mathbf{\Omega}_a$ are sampled together in one batch. We propose the bag-aggregation distillation loss including the intra-sample distillation loss $\mathcal{L}_{intra}$ and inter-sample distillation loss $\mathcal{L}_{inter}$.

To aggregate the representations within a bag into more compact embeddings, we minimize the following target function:

$$\min_{\theta_S} \mathcal{L} = \mathbb{E}_{\mathbf{x_i} \sim \mathbf{\Omega}_a} (L(f_\mathbf{S}(\mathbf{x_i}), f_\mathbf{T}(\mathbf{x}_a))), \tag{5}$$

where $L$ is a metric function to measure the distance between two embeddings – there are many metrics can be selected, such as cosine similarity, euclidean distance, *etc*. Here we use the normalized cosine similarity, *i.e.*, the contrastive loss commonly used in self-supervised learning to measure the distance between $\mathbf{x_i}$ and the anchor sample $\mathbf{x}_a$. The target function in Eq. 5 can be divided into two components:

$$\mathcal{L} = L(f_\mathbf{S}(\mathbf{t_1}(\mathbf{x}_a)), f_\mathbf{T}(\mathbf{t_2}(\mathbf{x}_a))) + \mathbb{E}_{\mathbf{x}_i \sim \mathbf{\Omega}_a \backslash \mathbf{x}_a} (L(f_\mathbf{S}(\mathbf{t_3}(\mathbf{x_i})), f_\mathbf{T}(\mathbf{t_2}(\mathbf{x}_a)))), \tag{6}$$

Three separate data augmentation operators $\mathbf{t}_1, \mathbf{t}_2, \mathbf{t}_3$ are randomly sampled from the same family of MoCo-v2 augmentations $\mathcal{T}$, which is also adopted in SEED(Fang et al., 2020) and DisCo (Gao et al., 2021). where the first item focuses on pulling different views (augmentations) of the same sample together, and the second item aims at pulling different samples that are within a same bag into more related ones. We term the first item as $\mathcal{L}_{intra}$ and the second item as $\mathcal{L}_{inter}$.

**Intra-Sample Distillation** The intra-sample distillation loss is a variant of conventional contrastive loss. Contrastive learning aims to learn representations by discriminating the positive key among negative samples. Given two augmented views $\mathbf{x}$ and $\mathbf{x}'$ of one input image, MoCo (Chen et al., 2020c) uses a online encoder $f_\mathrm{q}$ and a momentum encoder $f_\mathrm{k}$ to generate embeddings of the positive pairs: $q = f_\mathrm{q}(\mathbf{x})$, $k = f_\mathrm{k}(\mathbf{x}')$. The contrastive loss is defined as

$$\mathcal{L}_{contrast} = -\log \frac{\exp(\mathbf{q} \cdot \mathbf{k}^+ / \tau)}{\sum_{i=0}^N \exp(\mathbf{q} \cdot \mathbf{k_i} / \tau)}. \tag{7}$$

During distillation, we simply replace $f_\mathbf{q}$ and $f_\mathbf{k}$ by the student model $f_\mathbf{S}$ and teacher model $f_\mathbf{T}$, while weights of the teacher model $f_\mathbf{T}$ are pretrained and are not updated during distillation. The intra-sample distillation loss can be formulated as

$$\mathcal{L}_{intra} = -\log \frac{\exp(f_\mathbf{S}(\mathbf{t_1}(\mathbf{x}_a)) \cdot f_\mathbf{T}(\mathbf{t_2}(\mathbf{x}_a)) / \tau)}{\sum_{i=0}^N \exp(f_\mathbf{S}(\mathbf{t_1}(\mathbf{x}_a)) \cdot \mathbf{k_i}^- / \tau)}, \tag{8}$$

where $\tau$ is the temperature parameter. We select negative samples $\mathbf{k}^-$ in a memory bank, which is widely used in MoCo (He et al., 2020) and many subsequent contrastive learning methods. The memory bank is a queue of data embeddings and the queue size is much larger than a typical mini-batch size. After each forward iteration, items in the queue are progressively replaced by the current output of the teacher network.

**Inter-Sample Distillation** Given the anchor sample $\mathbf{x}_a$ and a positive sample $\mathbf{x}_p$ in the bag $\mathbf{\Omega}_a$, it is natural to map highly related samples to more similar representations. In other words, we want the

bag filled with related samples to be more compact. Inspired by Eq. 8, we define the inter-sample distillation loss as

$$\mathcal{L}_{inter} = -\log \frac{\exp(f_{\mathbf{S}}(\mathbf{t_3}(\mathbf{x}_p)) \cdot f_{\mathbf{T}}(\mathbf{t_2}(\mathbf{x}_a))/\tau)}{\sum_{i=0}^{N} \exp(f_{\mathbf{S}}(\mathbf{t_3}(\mathbf{x}_p)) \cdot \mathbf{k_i^-}/\tau)}. \tag{9}$$

The intra- and inter-sample distillation loss serve as different roles. The intra-sample distillation works like conventional distillation (Hinton et al., 2015; Romero et al., 2014), which aims at minimizing distances between outputs of the teacher and student model given the same input. However, the inter-sample distillation mainly focuses on transferring the data relation knowledge taking the bag-wise dataset as the carrier, which is obtained from the pretrained teacher model.

## 4 EXPERIMENTS

In this section, we evaluate the feature representations of the distilled student networks on several widely used benchmarks. We first report the performance on ImageNet under the linear evaluation and semi-supervised protocols. Then we conduct evaluation on several downstream tasks including object detection and instance segmentation, as well as some ablation studies to diagnose how each component and parameter affect the performance.

### 4.1 PRE-TRAINING DETAILS

**Pre-training of Teacher Model** Two models are used as teachers: ResNet-50 trained with MoCo-v2 (Chen et al., 2020c) for 800 epochs and ResNet-50×2 trained with SwAV for 400 epochs. The officially released weights [1] are used to initialize teacher models for fair comparisons.

**Self-supervised Distillation of Student Model** Two models are used as students: ResNet-18 and ResNet-34. Following the settings of MoCo in Chen et al. (2020c), we add a 2-layer MLP on top of the last averaged pooling layer to form a 128-d embedding vector. During distillation, the model is trained with the SGD optimizer with momentum 0.9 and weight decay 0.0001 for 200 epochs on ImageNet (Deng et al., 2009). The batch size and learning rate are set as 256 and 0.03 for 8 GPUs, which simply follow the hyper-parameter settings as in Chen et al. (2020c). The learning rate is decayed to 0 by a cosine scheduler. The CutMix used in Gidaris et al. (2021) and Xu et al. (2020b) is also applied to boost the performance. The temperature $\tau$ and the size of memory bank are set as 0.2 and 65,536 respectively. For the bagging strategy, we use K-nearest neighbors unless specified.

### 4.2 EXPERIMENTS ON IMAGENET

**KNN Classification** We evaluate representation of student model using nearest neighbor classifier. KNN classifier can evaluate the learned feature more directly without any parameter tuning. Following Caron et al. (2020); Abbasi Koohpayegani et al. (2020); Fang et al. (2020), we extract features from center-cropped images after the last averaged pooling layers. For convenient comparisons with other methods, we report the validation accuracy with 10 NN (we use the student model distilled from ResNet-50×2). As shown in Table 1, BINGO achieves 61.0% and 64.9% accuracies on ResNet-18/34 models, respectively, which outperforms previous methods significantly.

Table 1: KNN classification accuracy on ImageNet. We report the results on the validation set with 10 nearest neighbors. ResNet-50×2 is used as the teacher.

| Method | ResNet-18 | ResNet-34 |
|---|---|---|
| SEED (Fang et al., 2020) | 55.3 | 58.2 |
| BINGO | **61.0** | **64.9** |

**Linear Evaluation** In order to evaluate the performance of BINGO, we train a linear classifier upon the frozen representation, following the common evaluation protocol in Chen et al. (2020c). For fair comparisons, we use the same hyper-parameters as Fang et al. (2020); Gao et al. (2021) during linear evaluation stage. The classifier is trained for 100 epochs, using the SGD optimizer with 30 as initial learning rate. As shown in Table 2, BINGO outperform previous DisCo and SEED with the

---

[1]Checkpoints of teacher models can be downloaded from `https://github.com/facebookresearch/moco` and `https://github.com/facebookresearch/swav`.

Table 2: Linear classification accuracy on ImageNet over different student architectures. Note that when using R50×2 as the teacher, SEED distills for 800 epochs while DisCo and BINGO distill for 200 epochs. The numbers in brackets indicate the accuracies of teacher models. "T" denotes the teacher and "S" denotes the student.

| Method | S \ T | R-18 | | R-34 | |
|---|---|---|---|---|---|
| | | T-1 | T-5 | T-1 | T-5 |
| **Supervised** (Fang et al., 2020) | | 69.5 | - | 72.8 | - |
| MoCo-V2 (Baseline) (Fang et al., 2020) | | 52.5 | 77.0 | 57.4 | 81.6 |
| SEED (Fang et al., 2020) | R-50 (67.4) | 57.6 | 81.8 | 58.5 | 82.6 |
| DisCo (Gao et al., 2021) | R-50 (67.4) | 60.6 | 83.7 | 62.5 | 85.4 |
| BINGO | R-50 (67.4) | **61.4** | **84.3** | **63.5** | **85.7** |
| BINGO | R-50 (71.1) | **64.0** | **85.7** | **66.1** | **87.2** |
| SEED (Fang et al., 2020) | R-152 (74.1) | 59.5 | 83.3 | 62.7 | 85.8 |
| DisCo (Gao et al., 2021) | R-152 (74.1) | 65.5 | 86.7 | 68.1 | 88.6 |
| BINGO | R-152 (74.1) | **65.9** | **87.1** | **69.1** | **88.9** |
| SEED (Fang et al., 2020) | R50×2 (77.3) | 63.0 | 84.9 | 65.7 | 86.8 |
| DisCo (Gao et al., 2021) | R50×2 (77.3) | 65.2 | 86.8 | 67.6 | 88.6 |
| BINGO | R50×2 (77.3) | **65.5** | **87.0** | **68.9** | **89.0** |

Table 3: Transfer learning accuracy (%) on COCO detection.

| Method | Mask R-CNN, ResNet-18, Detection | | | | | | | | | | |
|---|---|---|---|---|---|---|---|---|---|---|---|
| | 1× schedule | | | | | | 2× schedule | | | | |
| | $AP^{bb}$ | $AP^{bb}_{50}$ | $AP^{bb}_{75}$ | $AP_S$ | $AP_M$ | $AP_L$ | $AP^{bb}$ | $AP^{bb}_{50}$ | $AP^{bb}_{75}$ | $AP_S$ | $AP_M$ | $AP_L$ |
| MoCo v2 | 31.3 | 50.0 | 33.5 | 16.5 | 33.1 | 41.1 | 34.4 | 53.9 | 37.0 | 18.9 | 36.8 | 45.5 |
| BINGO | **32.0** | **51.0** | **34.7** | **17.1** | **34.1** | **42.0** | **34.9** | **54.2** | **37.7** | **20.0** | **37.1** | **46.0** |

Table 4: Transfer learning accuracy (%) on COCO instance segmentation.

| Method | Mask R-CNN, ResNet-18, Instance Segmentation | | | | | | | | | | |
|---|---|---|---|---|---|---|---|---|---|---|---|
| | 1× schedule | | | | | | 2× schedule | | | | |
| | $AP^{mk}$ | $AP^{mk}_{50}$ | $AP^{mk}_{75}$ | $AP_S$ | $AP_M$ | $AP_L$ | $AP^{mk}$ | $AP^{mk}_{50}$ | $AP^{mk}_{75}$ | $AP_S$ | $AP_M$ | $AP_L$ |
| MoCo v2 | 28.8 | 47.2 | 30.6 | 12.2 | 29.7 | 42.7 | 31.5 | 51.1 | 33.6 | 14.1 | 32.9 | 46.9 |
| BINGO | **29.6** | **48.2** | **31.5** | **12.8** | **30.8** | **43.0** | **31.9** | **51.7** | **33.9** | **14.9** | **33.1** | **47.2** |

same teacher and student models. Note that SEED distills for 800 epochs while BINGO runs for 200 epochs with ResNet-50×2 teacher, which demonstrates the effectiveness of BINGO.

**Transfer to Object Detection and Instance Segmentation**    We evaluate the generalization ability of the student model on detection and instance segmentation tasks. The COCO dataset is used for evaluation. Following He et al. (2020), we use Mask R-CNN (He et al., 2017) for object detection and instance segmentation and fine-tune all the parameters of student model ResNet-18 end-to-end. As shown in Table 3 and Table 4, BINGO consistently outperforms models pretrained without distillation. The detection and segmentation results of ResNet-34 can be found in the appendix.

**Semi-supervised Classification**    Following previous works Chen et al. (2020a;b), we evaluate the proposed method by fine-tuning the student model ResNet-18 with 1% and 10% labeled data. We follow the training split settings as in Chen et al. (2020a) for fair comparisons. The network is fine-tuned for 60 epochs with SGD optimizer. The learning rate of the last randomly initialized fc layer is set as 10. As shown in Table 5, using ResNet-18 student model, BINGO obtains best accuracies with the same ResNet-50 and ResNet-152 teacher when using 1% and 10% labels, respectively.

**Transfer to CIFAR-10/CIFAR-100 classification**    Following the evaluation protocol in Fang et al. (2020); Gao et al. (2021), we assess the generalization of BINGO on the CIFAR-10/CIFAR-100 dataset. As shown in Table 6, compared with the previous state-of-the-art method DisCo (Gao et al., 2021), BINGO outperforms it by 1.5% and 3.2% respectively.

Table 5: Semi-supervised learning by fine-tuning $1\%$ and $10\%$ images on ImageNet using ResNet-18.

| Method | T | 1% labels | 10% labels |
|---|---|---|---|
| MoCo v2 baseline | - | 30.9 | 45.8 |
| Compress (Abbasi Koohpayegani et al., 2020) | R-50 (67.4) | 41.2 | 47.6 |
| SEED (Fang et al., 2020) | R-50 (67.4) | 39.1 | 50.2 |
| DisCo (Gao et al., 2021) | R-50 (67.4) | 39.2 | 50.1 |
| BINGO | R-50 (67.4) | **42.8** | **57.5** |
| Compress (Abbasi Koohpayegani et al., 2020) | R-152 (74.1) | - | - |
| SEED (Fang et al., 2020) | R-152 (74.1) | 44.3 | 54.8 |
| DisCo (Gao et al., 2021) | R-152 (74.1) | 47.1 | 54.7 |
| BINGO | R-152 (74.1) | **50.3** | **61.2** |
| BINGO | R-50×2 (77.3) | **48.2** | **60.2** |

Table 6: Linear classification accuracy on CIFAR-10/100 with ResNet-18.

| Method | T | CIFAR-10/100 |
|---|---|---|
| MoCo v2 baseline | - | 77.9/48.1 |
| SEED | R-50 (67.4) | 82.3/56.8 |
| DisCo | R-50 (67.4) | 85.3/63.3 |
| BINGO | R-50 (67.4) | **86.8/66.5** |

Table 7: Lower and Upper bound performance exploration via the bagging criterion.

| Bagging Criterion | Accuracy (%) |
|---|---|
| Random Initialized model | 46.6 |
| Supervised-pretrained model | 64.8 |
| Ground-truth labels | 65.8 |
| Self-supervised pretrained model | 64.0 |

### 4.3 ABLATION STUDY

In this section, we conduct detailed ablation studies to diagnose how each component affect the performance of the distilled model. Unless specified, all results in this section are based on ResNet-18, and distilled for 200 epochs.

**Impact of $k$ in K-nearest Neighbors** We inspect the influence of $k$ in K-nearest neighbors bagging strategy. As shown in Fig. 3, the results are relatively robust for a range of $k$ ($k$=1,5,10,20). In addition, we find that the classification accuracy decrease with $k = 10, 20$ compared with $k = 5$, because the noise is introduced when $k$ becomes large. However, the performance with a relative small $k = 1$ is no better than $k = 5$, we think the diversity is sacrificed when we only select the top-1 nearest neighborhood all the time.

**Lower and Upper Bound of The Proposed Method** As shown in Table 7, using data relation extracted from a random initialized model gives a poor performance of $46.6\%$, which can be a lower bound of our method. Then we try to explore the upper bound performance by bagging instances via a supervised-pretrained model, the performance gets an improvement of $0.8\%$ over using data relation extracted from the unsupervised pretrained teacher model. When we directly use the ground truth labels to bag instances, we get a highest upper bound performance, *i.e.*, $65.8\%$ Top-1 accuracy.

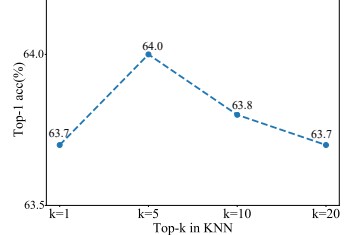

Figure 3: Top-1 accuracy with different $k$ in K-nearest neighbors.

**Impacts of Data-Relation and Teacher Parameters.** In our experiments, both the data relation and model parameters of teacher model are used to distill student model. We diagnose how each component affects the distillation performance. As shown in Table 8, model parameters represent whether to load the teacher model parameters into the teacher part in Fig. 2; data relation refers to the $\mathcal{L}_{inter}$ loss. Besides, $\mathcal{L}_{intra}$ is intrinsically a basic contrastive-learning paradigm on two views of one instance. no matter the teacher's parameters are loaded or not, using the the data relation from pretrained teacher model always gets better results than using data relation from online student model, which verifies the efficiency of transferring teacher's data relation to student model. Interestingly, we find that BINGO even gets good result only utilizing teacher's data relation (Row 3 of Table 8), which is about $10\%$ higher than model training without distillation.

Table 8: Effects of utilizing teacher's data-relation and teacher's pretrained weights. The column of Student Relation means that we bag data with features extracted from student model online and the column of Teacher Relation means that we bag data with features extracted from a pretrained teacher model. When teacher parameters are not used, we replace the pretrained teacher model as a momentum update of student model like He et al. (2020).

| Teacher Parameters | Student Relation | Teacher Relation | Accuracy |
|---|---|---|---|
| ✗ | ✗ | ✗ | 52.2 (w/o distillation) |
| ✗ | ✓ | ✗ | 57.2 |
| ✗ | ✗ | ✓ | 62.2 |
| ✓ | ✗ | ✗ | 62.0 |
| ✓ | ✓ | ✗ | 62.5 |
| ✓ | ✗ | ✓ | 64.0 |

Table 10: Top-1 accuracy of linear classification results on ImageNet with ResNet-34 under different epochs. ResNet-152 is used as teacher model. "T" denotes the teacher and "S" denotes the student.

| Method | S | T | Distillation Epochs | Top-1 Accuracy | Top-5 Accuracy |
|---|---|---|---|---|---|
| SEED | R-34 | R-152(74.1) | 200 | 62.7 | 85.8 |
| BINGO | R-34 | R-152(74.1) | 100 | 67.8 | 88.4 |
| DisCo | R-34 | R-152(74.1) | 200 | 68.1 | 88.6 |
| BINGO | R-34 | R-152(74.1) | 200 | 69.1 | 88.9 |

**Compare with Other Distillation Methods.** We now compare with several other distillation strategies to verify the effectiveness of our method. We compare with two distillation schemes: feature-based distillation method and relation-based distillation, which is termed as KD and RKD, respectively. Feature-based distillation method aims at minimizing $l2$-distance of teacher & student's embeddings. Relation-based distillation method aims at minimizing the difference between inter-sample-similarity graph obtained from teacher and student model. As shown in Table 9, BINGO outperforms all theses alternative methods.

Table 9: Top-1 accuracy of linear classification results on ImageNet using different distillation methods on ResNet-18 student model (ResNet-50 is used as teacher model)

| Method | Top-1 |
|---|---|
| MoCo-V2 baseline (Gao et al., 2021) | 52.2 |
| MoCo-V2 + KD (Fang et al., 2020) | 55.3 |
| MoCo-V2 + RKD | 61.6 |
| DisCo + KD (Gao et al., 2021) | 60.6 |
| DisCo + RKD (Gao et al., 2021) | 60.6 |
| BINGO | **64.0** |

**Computational Complexity.** As shown in Fig. 2, the batch size is doubled during each forward propagation. The computation cost is increased compared with SEED (Fang et al., 2020) and MoCo-v2 (Chen et al., 2020c). As for DisCo (Gao et al., 2021), the computational complexity is also increased due to the multiple forward propagation for one sample: once for the mean student, twice for the online student and twice for the teacher model. The total number of forward propagation is 5, $2.5\times$ bigger than SEED and MoCo-v2. For the above analysis, BINGO has a similar computation cost with DisCo. To compare with SEED and DisCo under the same cost, we distill ResNet-34 from ResNet-152 for 100 and 200 epochs respectively. We compare results of 100 epochs with SEED and 200 epochs with DisCo. The results are shown in Table 10 and BINGO still shows significantly better performance than the two compared methods with the same training cost.

## 5 CONCLUSIONS

This paper proposes a new self-supervised distillation method, named BINGO, which bags related instances by matching embeddings of the teacher. With the instance-wise dataset mapped into a bag-wise one, the new dataset can be applied to the distillation process for small models. The knowledge which represents the relation of bagged instances can be transferred by aggregating the bag, including inter-sample and intra-sample aggregation. Our BINGO follows a relation-guided principle, which shows stronger effectiveness than previous relation-agnostic methods. The proposed relation-based distillation is a general strategy for improving unsupervised representation, and we hope it would shed light on new directions for unsupervised learning.

## ACKNOWLEDGEMENT

This work was supported in part by the National Natural Science Foundation of China under Grant 61932022, Grant 61720106001, Grant 61971285, and in part by the Program of Shanghai Science and Technology Innovation Project under Grant 20511100100.

## REPRODUCIBILITY STATEMENT

We will release the source code at `https://github.com/haohang96/bingo` to guarantee the reproducibility of this paper.

## ETHICS STATEMENT

The proposed approach seeks to improve the performance of unsupervised learning methods for small scale models. This will improve the application of unsupervised learning methods in more realistic computing resource-constrained scenarios. As far as we know, our method does not violate any ethical requirements.

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

## A  APPENDIX

### A.1  RESULTS OF K-MEANS BAGGING STRATEGY

Table 11: Top-1 accuracy with different cluster numbers $\mathbf{C}$ in K-means clustering.

| Number of Clusters ($\mathbf{C}$) | Accuracy |
| --- | --- |
| 5000 | 62.7 |
| 10000 | 63.5 |
| 20000 | 63.8 |
| 50000 | 63.6 |

We additionally evaluate the performance of using K-means clustering as the bagging strategy according to Eq. 2 in the main text. Given the pseudo-label $\mathbf{q} = \{\mathbf{q_1}, \mathbf{q_2}, ..., \mathbf{q_N}\}$ and the anchor instance $\mathbf{x}_a$, the bag associated with $\mathbf{x}_a$ is defined as

$$\mathbf{\Omega}_a = \{\mathbf{i} \mid \mathbf{q_i} = \mathbf{q}_a, \mathbf{i} = 1, 2, ..., N\}. \tag{10}$$

For implementation, ResNet-18 and ResNet-50 are used as the student and teacher model respectively. We evaluate the linear classification accuracy of the student model on ImageNet-1K. We study various cluster numbers $\mathbf{C}$ as shown in Tab. 11. We find that a bigger cluster number can bring better results than a smaller one. Noting that the linear classification accuracy of bagging with K-nearest neighbors (where $k$=5) is slightly better than bagging via K-means clustering (with $\mathbf{C}$=20000), *i.e.* $64.0\%$ *vs.* $63.8\%$. Moreover, bagging with KNN is more convenient to implement, so we choose the KNN-based bagging strategy in implementation.

### A.2  OBJECT DETECTION AND INSTANCE SEGMENTATION OF RESNET-34

We further evaluate the generalization ability of one more student model, *i.e.* ResNet-34, on object detection and instance segmentation tasks. The COCO (Lin et al., 2014) dataset is used for evaluation. Following He et al. (2020), we use Mask R-CNN (He et al., 2017) for object detection and instance segmentation, and fine-tune parameters of the student model , *i.e.* ResNet-34 distilled from the ResNet-152 teacher model. As shown in Table 12, BINGO consistently outperforms models pretrained with no distillation, SEED Fang et al. (2020) and DisCo Gao et al. (2021).

Table 12: Transfer learning performance on COCO Lin et al. (2014) object detection and instance discrimination with ResNet-34 distilled from ResNet-152. "bb" denotes bounding box and "mk" denotes mask.

| Method | Mask R-CNN, ResNet-34 | | | | | | | | | | | |
|---|---|---|---|---|---|---|---|---|---|---|---|---|
| | Object Detection | | | | | | Instance Discrimination | | | | | |
| | $AP^{bb}$ | $AP^{bb}_{50}$ | $AP^{bb}_{75}$ | $AP_S$ | $AP_M$ | $AP_L$ | $AP^{mk}$ | $AP^{mk}_{50}$ | $AP^{mk}_{75}$ | $AP_S$ | $AP_M$ | $AP_L$ |
| MoCo v2 | 38.1 | 56.8 | 40.7 | - | - | - | 33.0 | 53.2 | 35.3 | - | - | - |
| SEED Fang et al. (2020) | 38.4 | 57.0 | 41.0 | - | - | - | 33.3 | 53.7 | 35.3 | - | - | - |
| DisCo Gao et al. (2021) | 39.4 | 58.7 | 42.7 | - | - | - | 34.4 | 55.4 | 36.7 | - | - | - |
| BINGO | **39.9** | **59.4** | **43.5** | 22.8 | 43.3 | 52.1 | **35.7** | **56.5** | **38.2** | 16.8 | 37.9 | 51.6 |

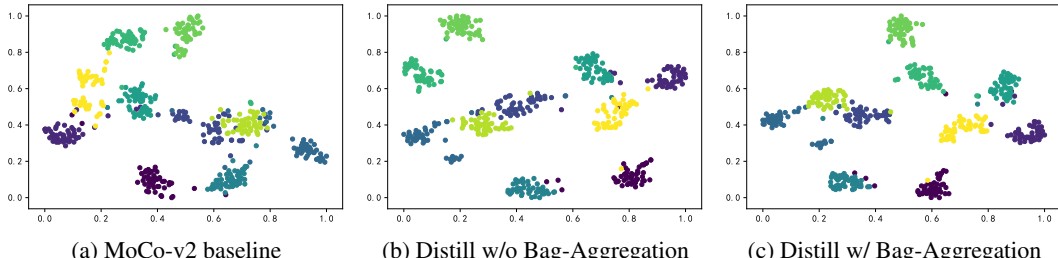

(a) MoCo-v2 baseline    (b) Distill w/o Bag-Aggregation    (c) Distill w/ Bag-Aggregation

Figure 4: *t-sne* visualization of student's representations pretrained with the MoCo-v2 baseline (a), and distilled with (b) and without bag aggregation (c).

## A.3 ANALYSIS AND DISCUSSIONS

We now inspect what the student learns during the distillation. Firstly we compute the average distance between anchor sample $\mathbf{x}_a$ and its positive samples $\mathbf{x_p}$ in a bag $\mathbf{\Omega}_a$ over the whole dataset:

$$\mathbf{BagDis} = \mathop{\mathbb{E}}_{\mathbf{x}_a \sim \boldsymbol{X}} \mathop{\mathbb{E}}_{\mathbf{x}_p \sim \boldsymbol{\Omega}_a} ||(f_{\mathbf{S}}(\mathbf{x}_a) - f_{\mathbf{S}}(\mathbf{x}_p))||_2^2 \qquad (11)$$

According to Eq. 11, we compute the averaged distance in the bag using distilled student model. As shown in Table 13, the averaged distance in a bag is smallest when the student model is distilled with bag-aggregation loss. We also compute the intra-class distance among all intra-class pairwise samples. As shown in Table 14, the proposed method also aggregate the bag of labels with the same ground truth labels on the unseen validation set.

Table 13: Averaged distance between anchor and positive samples in the same bag

| Method | Distance |
|---|---|
| MoCo-V2 baseline | 0.38 |
| Distill w/o Bag-Aggregation | 0.36 |
| Distill w/ Bag-Aggregation | 0.32 |

Table 14: Averaged intra-class distance on ImageNet validation set

| Method | Distance |
|---|---|
| MoCo-V2 baseline | 0.88 |
| Distill w/o Bag-Aggregation | 0.72 |
| Distill w/ Bag-Aggregation | 0.65 |

We visualize the last embedding feature to understanding the aggregating properties of the proposed method. 10 classes are randomly selected from validation set. We provide the *t-sne* visualization of the student features. As shown in Fig. 4, the same color denotes features with the same label. It can be seen that BINGO gets more compact representations compared with models without distillation or distilling without pulling related samples in a bag.

