# OpenReview forum: "Bag of Instances Aggregation Boosts Self-supervised Distillation"
_ICLR.cc/2022/Conference — ICLR 2022 Poster_

### Official Review · Reviewer_HSTU · 2021-11-02

**Correctness:** 3
**Technical Novelty And Significance:** 4
**Empirical Novelty And Significance:** 3
**Recommendation:** 6
**Confidence:** 5

**Main Review:**

The overall idea to novel to me, the idea of using pretrained teacher to aggregate similar samples is the main contribution and difference compared with previous works. The ablation study has well demonstrated the effectiveness of each component, but I'm a bit concerned about the main result since the comparison seems not fair with the previous method.

From my point of view, the experiment is not quite fair to compare with the previous method.

1. From Figure 2, it's clear to see that both X_{a} and X_{p} have to be passed into the student network, which means the batch size is doubled compared with SEED and DisCo. From my experience, the training will be at least 50% slower. On the other hand, increasing the number of positive samples can constantly improve the accuracy in contrastive learning. I'm looking forward to more experiments under the same training costs.

2. In Table 1 (first half result), the teacher of SEED and DisCo has an accuracy of 67.4%, but BINGO is trained with a better teacher (71.1%), which is not a fair comparison. I'm looking for the result with the same teacher. (under the same training cost).

3. Table 3 and Table 4 only report the result of ResNet18 and compare it with the MoCo baseline. The distilled network has a better performance than the baseline method is not a surprising phenomenon. I'm looking forward to more comparisons between BINGO, SEED, and DisCO. (under the same training cost).



**Summary Of The Paper:**

This paper proposes a new knowledge distillation framework for unsupervised representation learning. Unlike the previous distillation method, where the knowledge transfer only happens between the same image, BINGO utilized a pretrained teacher model to aggregate similar samples into bags. The goal is to aggregate compact representations over the student with respect to instances in a bag.


**Summary Of The Review:**

The paper is well written and easy to follow. As I have mentioned, I'm a bit concerned about the main result. Interestingly, I have reviewed this paper in another venue, but my questions are still not addressed well. However, I would like to increase my score if they can be handled well here.

---

> ### Author Response · Authors · 2021-11-21
> **Thanks and Response to Reviewer HSTU (1)**
>
> We sincerely thank the reviewer for your constructed advice and hope the provided clarifications could clear your concerns.
>
> > Q1: From Figure 2, it's clear to see that both X_{a} and X_{p} have to be passed into the student network, which means the batch size is doubled compared with SEED and DisCo. From my experience, the training will be at least 50% slower. On the other hand, increasing the number of positive samples can constantly improve the accuracy in contrastive learning. I'm looking forward to more experiments under the same training costs.
>
> **A1**: We detailedly analyze the computation cost as follows. DisCo forwards 5 views of one image in each iteration as described in Fig. 2 of [R1], i.e., 1 for the mean student, 2 for the online student, and 2 for the teacher model. This means that the actual batch size in DisCo is 2.5$\times$ of that in SEED and MoCo-v2. Though BINGO doubles the batch size compared with SEED but has a similar computation cost compared with DisCo.
>
> Compared with SEED, our method actually increases the cost per epoch, but still achieves better results with half epochs. To compare BINGO with SEED and DisCo with the same computation cost, we implement BINGO based on the ResNet-34 student and ResNet-152 teacher. We run BINGO for 100 epochs and 200 epochs respectively, which ensures the computation cost is similar with SEED and DisCo. The results are shown in the below tables:
>
> | Method | Student | Distill Epochs | Teacher      | Top-1 Acc | Top-5 Acc |
> | ------ | ------- | -------------- | ------------ | --------- | --------- |
> | SEED   | R-34    | 200            | R-152 (74.1) | 62.7      | 85.8      |
> | BINGO  | R-34    | 100            | R-152 (74.1) | 67.8      | 88.4      |
>
> | Method | Student | Distill Epochs | Teacher      | Top-1 Acc | Top-5 Acc |
> | ------ | ------- | -------------- | ------------ | --------- | --------- |
> | DisCo  | R-34    | 200            | R-152 (74.1) | 68.1      | 88.6      |
> | BINGO  | R-34    | 200            | R-152 (74.1) | 69.1      | 88.9      |
>
> In table 1 of initial submission, with the ResNet-50$\times$2 teacher, SEED distills for 800 epochs, while DisCo and BINGO distill for 200 epochs and take the smaller cost with SEED. Our method still shows superior performance than both SEED and DisCo.
>
>
>
> > Q2: In Table 1 (first half result), the teacher of SEED and DisCo has an accuracy of 67.4%, but BINGO is trained with a better teacher (71.1%), which is not a fair comparison. I'm looking for the result with the same teacher. (under the same training cost).
>
> **A2**: Both 67.4%- and 71.1%-accuracy checkpoints of the ResNet-50 teacher model can be directly downloaded from the [MoCo github repository](https://github.com/facebookresearch/moco). We simply choose a better-performed model as the teacher for distillation, and also choose ResNet-50$\times$2 with the same accuracy as the teacher for fair comparison in the original submission. We also supplement the results with the same ResNet-50 teacher with 67.4% accuracy and ResNet-152 teacher with 74.1% accuracy as follows and the advantage of our method is still evident.
>
> | Method | Teacher     |   R-18   |   R-34   |
> | ------ | ----------- | :------: | :------: |
> | SEED   | R-50 (67.4) |   57.6   |   58.5   |
> | DisCo  | R-50 (67.4) |   60.6   |   62.5   |
> | BINGO  | R-50 (67.4) | **61.4** | **63.5** |
> | BINGO  | R-50 (71.1) | **64.0** | **66.1** |
>
> | Method | Teacher      |   R-18   |   R-34   |
> | ------ | ------------ | :------: | :------: |
> | SEED   | R-152 (74.1) |   59.5   |   62.7   |
> | DisCo  | R-152 (74.1) |   65.5   |   68.1   |
> | BINGO  | R-152 (74.1) | **65.9** | **69.1** |
>
> Note that in table 1, with the ResNet-50$\times$2 teacher, SEED distills for 800 epochs, while DisCo and BINGO distill for 200 epochs and take the smaller cost with SEED. Our method still shows superior performance than both SEED and DisCo.

---

> ### Author Response · Authors · 2021-11-21
> **Thanks and Response to Reviewer HSTU (2)**
>
> > Q3: Table 3 and Table 4 only report the result of ResNet18 and compare it with the MoCo baseline. The distilled network has a better performance than the baseline method is not a surprising phenomenon. I'm looking forward to more comparisons between BINGO, SEED, and DisCO. (under the same training cost).
>
> **A3**: Thanks for your advice. DisCo only evaluates object detection and instance segmentation tasks based on ResNet-34 network. For a fair comparison, we implement BINGO based on the ResNet-34 student with the ResNet-152 teacher model the same as that in DisCo and SEED. The results of object detection and instance segmentation on COCO are shown in the below table:
>
> | Method | Student | Teacher | Pretraining Epochs | $AP^{bb}$ | $AP^{bb}_{50}$ | $AP^{bb}_{75}$ | $AP^{mk}$ | $AP^{mk}_{50}$ | $AP^{mk}_{75}$ |
> | ------ | ------- | ------- | ------------------ | --------- | -------------- | -------------- | --------- | -------------- | -------------- |
> | SEED   | R-34    | R-152   | 200                | 38.4      | 57.0           | 41.0           | 33.3      | 53.7           | 35.3           |
> | BINGO  | R-34    | R-152   | 100                | **38.7**  | **58.2**       | **42.1**       | **34.7**  | **55.0**       | **37.1**       |
>
> | Method | Student | Teacher | Pretraining Epochs | $AP^{bb}$ | $AP^{bb}_{50}$ | $AP^{bb}_{75}$ | $AP^{mk}$ | $AP^{mk}_{50}$ | $AP^{mk}_{75}$ |
> | ------ | ------- | ------- | ------------------ | --------- | -------------- | -------------- | --------- | -------------- | -------------- |
> | DisCo  | R-34    | R-152   | 200                | 39.4      | 58.7           | 42.7           | 34.4      | 55.4           | 36.7           |
> | BINGO  | R-34    | R-152   | 200                | **39.9**  | **59.4**       | **43.5**       | **35.7**  | **56.5**       | **38.2**       |
>
>
>
> [R1] Gao, Yuting, et al. "DisCo: Remedy Self-supervised Learning on Lightweight Models with Distilled Contrastive Learning." *arXiv preprint arXiv:2104.09124* (2021).

---

> > ### Comment · Reviewer_HSTU · 2021-12-08
> > **Thanks for your reply.**
> >
> > The experimental results seem ok to me. I decide to change my score a bit.

---

> > > ### Author Response · Authors · 2021-12-08
> > > **Thanks for Your Response**
> > >
> > > Many thanks for your quick response. Your suggestions help us a lot in improving the paper and the updates will be included in our final version.

---

> ### Author Response · Authors · 2021-11-26
> **Thanks and Looking Forward to Your Reply**
>
> Dear Reviewer,
>
> we would appreciate your review work which helps us a lot in improving our paper. We wonder if there is any additional question and would like to answer at any time. If our response has cleared your concerns, could you please consider raising the rating?
>
> Many thanks for your comments again.

---

> > ### Author Response · Authors · 2021-12-08
> > **A Kind Reminder and Looking Forward to Your Reply**
> >
> > Dear Reviewer,
> > many thanks for all your efforts in the review work. We still sincerely look forward to your response and wonder if our response has addressed your questions. We would appreciate your time so much and your reply is very important to us. Sorry for bothering you.

---

### Official Review · Reviewer_Xmuw · 2021-11-02

**Correctness:** 4
**Technical Novelty And Significance:** 3
**Empirical Novelty And Significance:** 3
**Recommendation:** 6
**Confidence:** 4

**Main Review:**

Some suggestion regards the writing:

- Wrong reference for Compress in the beginning of third paragraph:
"Compress (Fang et al., 2021) and SEED (Fang et al., 2021) are two typical methods for unsupervised
distillation"
-  large quantity of exemplars -> large number of exemplars
- drops dramatically comparing with its supervised counterpart -> compared with its
- termed as BINGO, which is short -> termed BINGO
- targets at transferring the relationship -> targets transferring ...
- while DisCo and BINGO distills for 200 epochs -> distill for
- all theses alternative methods -> these
- the proposed method also aggregate the bag -> aggregates
- using the the data relation from pretrained teacher -> duplicate ``the''


======================================================================================

Main Review:

This paper is well-established upon the previous self-supervised distillation efforts that aims to train stronger small models via the self-supervised fashion. Previous works like Compress and SEED mainly focus on the instance-level contrastive learning, where all samples are treated equivalently as negative samples without relation-distillation. The key contribution of the BINGO is the propose of inter-sample loss (eq. 9) that additionally push sample embeddings with identical pseudo-labels closer.  Intuitively, the cluster -> label assignment measure is similar to SWAV, where the Sinkhorn algorithm is instead adopted and the representation of target model itself is leveraged for clustering.

The major concern from my side is the technical novelty of the BINGO and the proposed relation-distillation: it feels to me that the main reason that inter-sample distillation works is its mimicking the representations from other categories. How is the relation-distillation reflected in the inter-sample distillation loss? As it's not mimicking any categorical distributions and I'm not quite sure that can be viewed as relation across samples.

======================================================================================

Some Questions Regards the Implementation & Evaluations details:

- The fact that K-Means clustering method works well is mostly because the features from Teacher model is kept frozen. A recent work in [1] proposes to use the trainable Teacher and Sinkhorn instead as clustering algorithm. This deserves further discussion to differentiate BINGO with it.

- Table 1: results of BINGO seems to have used a stronger ResNet-50 as the Teacher (71.1% v.s. 67.4% from SEED and DisCo), what is the reason for this inconsistent comparisons?

- Transfer learning results (detection/segmentation) only include the MoCo-v2 results, how about the numbers comparing with DisCo and SEED?

- It seems that the number of clustering centers may lead to pretty diverse results, what is the intuition in selecting the optimal K?

- According to SEED & SWAV, they uses multi-view strategy, is this adopted in BINGO? As I never see any description about it.


[1] Unsupervised Representation Transfer for Small Networks: I Believe I Can Distill On-the-Fly, Hee Min Choi, Hyoa Kang, and Dokwan Oh, Samsung Advanced Institute of Technology, Suwon, South Korea, NeurIPS 2021



**Summary Of The Paper:**

This paper presents a new self-supervised distillation learning schema which aims to address the ``low-relation'' distillation objective across instances in priori arts. In particular, the developed bagging instance technique cluster the offline Teacher representations into $K$ bags, assigning sample categorical labels. The overall training objective of BINGO is composed of the intra-sample distillation term, which retains the contrastive learning principle; and the inter-sample distillation term that select pseudo-positive samples from the clustering labels for further contrastive learning. Quantitative experiments on ImageNet classification task, transferring learning and semi-supervised learning task validate the effectiveness of proposed method.

**Summary Of The Review:**

- The paper is well-constructed with clear motivation and comprehensive quantitative experiments.
- Experiments show that the proposed BINGO learning schema is valid and bring incremental improvement over previous SEED and Compress.
- My main concern is the technical contribution of BINGO somehow overlaps with [1] as previously mentioned. And some experimental settings are not consistent or clearly explained.
I'd like to adjust my initial ratings after reading the response afterwards.

---

> ### Author Response · Authors · 2021-11-21
> **Thanks and Response to Reviewer Xmuw (1)**
>
> Thank the reviewer for your detailed comments. We answer your questions as follows and hope the response could clear your concerns.
>
> First of all, we would like to provide a clarification on our method to clear misunderstanding as possible. Data relation knowledge proposed in our method represents the connection between **different instances with similar features**, namely bagging similar instances together. It is totally different from previous methods, e.g. SEED[R1] or SwAV[R2], which align distributions of **one instance's different views** on other categories. And in our study, substantial proofs demonstrate our newly proposed relation knowledge has much stronger ability to boost small model's leaning. We sincerely hope this clarification clears the misunderstanding about the two different types of distillation knowledge.
>
> > Q1: Intuitively, the cluster -> label assignment measure is similar to SwAV, where the Sinkhorn algorithm is instead adopted and the representation of target model itself is leveraged for clustering.
>
> **A1**: SEED[R1] shares a similar methodology as SwAV[R2], *i.e.*, using cross-entropy loss to align feature distribution between views of **the same instance**. Our method aims at a new perspective by using a bag-wise dataset obtained from the teacher model to represent data relation between **different instances**, which shows evident advantages of small model learning representations  on both ImageNet classification and several downstream tasks. In addition, the proposed bag-wise relation knowledge could be integrated with SwAV-based distillation methods, which we leave as future work.
>
>
> > Q2: The major concern from my side is the technical novelty of the BINGO and the proposed relation-distillation: it feels to me that the main reason that inter-sample distillation works is its mimicking the representations from other categories. How is the relation-distillation reflected in the inter-sample distillation loss? As it's not mimicking any categorical distributions and I'm not quite sure that can be viewed as relation across samples.
>
> **A2**: As mentioned in the above explanation, we propose a new type of data relation knowledge which exists among **similar instances**, different from mimicking distributions of a same instance's views on **different categories** in previous methods. The proposed relation-guided method is novel as Reviewer HSTU recognizes and shows much stronger ability of learning representations with small models than previous categorical-distribution methods.
>
> > Q3: The fact that K-Means clustering method works well is mostly because the features from Teacher model is kept frozen. A recent work in [1] proposes to use the trainable Teacher and Sinkhorn instead as clustering algorithm. This deserves further discussion to differentiate BINGO with it.
>
> **A3**: Sorry for this missing work, but when we submitted the paper to ICLR 2022, [1] was not yet released. We will cite [1] and add discussion in our revised paper.

---

> > ### Comment · Reviewer_Xmuw · 2021-11-22
> > **Thanks for the Good Clarification**
> >
> > I read through the response from the authors and it addresses most of my previously raised concerns and I'm glad to raise my initial rating. In addition to that, I strongly encourage the authors to further include all the updates in the future version.

---

> > > ### Author Response · Authors · 2021-11-23
> > > **Thanks for Your Response**
> > >
> > > Thank you so much for your kind response and time for the review work. We would update these changes in our future version.

---

> ### Author Response · Authors · 2021-11-21
> **Thanks and Response to Reviewer Xmuw (2)**
>
> > Q4: Table 1: results of BINGO seems to have used a stronger ResNet-50 as the Teacher (71.1% v.s. 67.4% from SEED and DisCo), what is the reason for this inconsistent comparisons?
>
> **A4**: Both 67.4%- and 71.1%-accuracy checkpoints of the ResNet-50 teacher model can be directly downloaded from the [MoCo github repository](https://github.com/facebookresearch/moco). We simply choose a better-performing model as the teacher for distillation, and also choose ResNet-50$\times$2 with the same accuracy as the teacher for fair comparison in the original submission. We also supplement the results with the same ResNet-50 teacher with 67.4% accuracy and ResNet-152 teacher with 74.1% accuracy as follows and the advantage of our method is still evident.
>
> | Method | Teacher     |   R-18   |   R-34   |
> | ------ | ----------- | :------: | :------: |
> | SEED   | R-50 (67.4) |   57.6   |   58.5   |
> | DisCo  | R-50 (67.4) |   60.6   |   62.5   |
> | BINGO  | R-50 (67.4) | **61.4** | **63.5** |
> | BINGO  | R-50 (71.1) | **64.0** | **66.1** |
>
> | Method | Teacher      |   R-18   |   R-34   |
> | ------ | ------------ | :------: | :------: |
> | SEED   | R-152 (74.1) |   59.5   |   62.7   |
> | DisCo  | R-152 (74.1) |   65.5   |   68.1   |
> | BINGO  | R-152 (74.1) | **65.9** | **69.1** |
>
>
> > Q5: Detection / Segmentation results compared with SEED and DisCo
>
> **A5**: Thanks for your advice. DisCo only evaluates object detection and instance segmentation tasks based on the ResNet-34 network. For a fair comparison, we implement BINGO based on the ResNet-34 student with the ResNet-152 teacher model the same as that in DisCo and SEED. The results of object detection and instance segmentation on COCO are shown in the below table:
>
> | Method | Student |   Teacher    | $AP^{bb}$ | $AP^{bb}_{50}$ | $AP^{bb}_{75}$ | $AP^{mk}$ | $AP^{mk}_{50}$ | $AP^{mk}_{75}$ |
> | ------ | :-----: | :----------: | --------- | -------------- | -------------- | --------- | -------------- | -------------- |
> | SEED   |  R-34   | R-152 (74.1) | 38.4      | 57.0           | 41.0           | 33.3      | 53.7           | 35.3           |
> | DisCo  |  R-34   | R-152 (74.1) | 39.4      | 58.7           | 42.7           | 34.4      | 55.4           | 36.7           |
> | BINGO  |  R-34   | R-152 (74.1) | **39.9**  | **59.4**       | **43.5**       | **35.7**  | **56.5**       | **38.2**       |
>
> > Q6: What is the intuition in selecting the optimal K
>
> **A6**: We have studied the impact of different $k$ in Fig. 3 of our initial submission. As shown in Fig. 3, $k=5$ obtains the best performance. We deduce too small $k$ will sacrifice the diversity of data relation, but too large $k$ will introduce more noise for learning.
>
> > Q7: Multi-view strategy
>
> **A7**: We use the standard configuration of MoCo-v2 but do not use the multi-view (multi-crop) augmentation strategy for efficiency, which can evidently improve the final performance but brings additional computation cost as used and described in SEED and SwAV. Our BINGO can outperform SEED significantly even without this multi-crop augmentation.
>
>
>
> [R1] Fang, Zhiyuan, et al. "SEED: Self-supervised Distillation For Visual Representation." *ICLR*, 2020.
>
> [R2] Caron, Mathilde, et al. "Unsupervised Learning of Visual Features by Contrasting Cluster Assignments." *Thirty-fourth Conference on Neural Information Processing Systems (NeurIPS)*. 2020.

---

### Official Review · Reviewer_dQqV · 2021-11-02

**Correctness:** 4
**Technical Novelty And Significance:** 3
**Empirical Novelty And Significance:** 2
**Recommendation:** 6
**Confidence:** 5

**Main Review:**

The paper tackles an interesting problem that is under-studied. It does tackle the problem following the same approach (a variant of KD applied to this problem). It presents clear novelties respect to prior art, but largely follows prior art in terms of the approach. The explanations are very clear and the paper is well written. Some related references could be added (see bellow), but overall it is complete. The experimental results seem a bit less complete for a number of reasons, although the results look to some extent promising.

I have two main comments: methodology/story and experiments.

In terms of the methodology, the approach seems to be pulling the k most similar examples together. Obviously, the model capacity is a problem and forcing the model to distinguish between similar instances is challenging. This is a softer approach in which semantically similar instances are grouped together and thus the whole learning is less ambitious (I feel one related work not included is "Khosla et al., Supervised Contrastive Learning, NeurIPS 2020", which targets a similar goal but uses the supervised learning labels). The question is how this "lack of modelling ambition" affects downstream tasks. When I was reading the methodology I thought this setting would be good for ImageNet (in the end you'll "cluster" different classes together), but maybe not so good for other downstream applications.

This takes me to the limitations of the experimental validation. Right now there is a very weak comparison on COCO, where competing methods are not included. There's no comparison on PASCAL or instance segmentation, etc.

Other comparison that is, in my opinion, not very conclusive is the semi supervised learning on imagenet. Other works (e.g. DisCo) show a graph of performance vs teacher. The largest teacher yields better performance than the one shown here, but I didn't see a clarification of which teacher is used.

Finally, for being a method for SSL on low-compute models, it focuses on ResNet18. Is the MobileNet family not considered for example? both DisCo and SEED do use MobileNetV3 and EfficientNet for example.


Small issues:
The related work focuses more on pretext tasks than on contrastive method. This could do with some rebalancing.
I'd suggest citing the conference paper rather than the arxiv version
Why on Table 1, the first two lines of the penultimate block have a different teacher performance than the 3rd line?
There's a typo in page 6: DINGO -> BINGO
Table 7, 3rd line, looks like a very weird result. Are you sure of that one? Do you have any explanation? or maybe I misunderstood it?
Some references
Khosla et al., Supervised Contrastive Learning, NeurIPS 2020 (referred to above)
Xu et al., Knowledge Distillation Meets Self-Supervision, ECCV'20 (basically SEED is this paper minus the supervised learning bit)
Dwibedi et al., With a Little Help from My Friends: Nearest-Neighbor Contrastive Learning of Visual Representations, ICCV'21


**Summary Of The Paper:**

The paper tackles the problem of training a self-supervised model using low(er)-compute models. Following the literature, the authors present a variant of distillation that achieves better performance than competing distillation methods. The base algorithm is based on contrastive learning (MoCoV2). The variant uses the teacher to find, for each instance in the training set, the k closest examples within the training set. Then, it uses two augmentations of the training instance as a positive pair (as customary for contrastive), but then also uses an example sampled from the k nearest to construct another positive pair. One element of each pair is passed through the teacher and another through the student, and each pair generates a loss.

**Summary Of The Review:**

While the method is clear and has an interesting premise, I am left wondering if the rationale behind the methodology is the right one. Experimental evidence is not very strong, and there are some clear aspects lacking in the evaluation. The most important issues are the lack of proper validation on downstream tasks, and the lack of variety of student architectures. (the score of 2 on "Empirical Novelty And Significance" is due to lack of completeness of experiments but can be raised with stronger evidence).

(updated to 6 after rebuttal)

---

> ### Author Response · Authors · 2021-11-21
> **Thanks and Response to Reviewer dQqV (1)**
>
> we sincerely thank you for your efforts on the review work and your detailed comments. We answer your questions as follows and hope the response could clear your concerns.
>
> > Q1: I feel one related work not included is "Khosla et al., Supervised Contrastive Learning, NeurIPS 2020", which targets a similar goal but uses the supervised learning labels.
>
> **A1**: Thank you for reminding. We will supplement this related work and demonstrate the relation difference between the two as follows. Both [R1] and our BINGO share a similar concept of constructing data relation and find this knowledge is very useful for representation learning. However, the goals and methodologies are quite different. [R1] focuses on the supervised scenario and performs contrastive learning using more positive samples with the same category labels. For the unsupervised task in BINGO, it is much more difficult to perform data relation construction. We propose a very interesting method by bagging feature-similar instances and aggregating the bag. We prove that transferring this relation knowledge between models can significantly strengthen small models' representation learning.
>
> > Q2: The question is how this "lack of modeling ambition" affects downstream tasks. When I was reading the methodology I thought this setting would be good for ImageNet (in the end you'll "cluster" different classes together), but maybe not so good for other downstream applications.
>
> **A2**: Good question! We demonstrate the benefits that our method could bring to downstream tasks from two main perspectives. One is from distillation, i.e., representation learning ability, where the student network learns better representations by mimicking the teacher's output, which is also proved to improve downstream tasks in both our work and previous methods SEED[R2] and DisCo[R3].
>
> The other one is from data relation. In contrastive learning, each image is treated as a separate class. The positive sample pairs are generated by random augmentations of one image, and all the other images are treated as negative samples. However, the semantically similar images are not connected explicitly, where they are on the contrary treated as negative ones and pushed away. This manner is harmful to the network learning an accurate data distribution.
>
> By introducing data relation obtained from teacher network in BINGO, the student model can learn more general representations and can easily fit the data distribution.
>
>
> > Q3: This takes me to the limitations of the experimental validation.
>
> **A3**: Thanks for your advice. DisCo only evaluates object detection and instance segmentation tasks based on ResNet-34 network. For a fair comparison, we implement BINGO based on the ResNet-34 student with the ResNet-152 teacher model the same as that in DisCo and SEED. The results of object detection ($AP^{bb}$ $AP_{50}^{bb}$, $AP_{75}^{bb}$) and instance segmentation ($AP^{mk}$, $AP_{50}^{mk}$, $AP_{75}^{mk}$) on COCO are shown in the below table:
>
> | Method | Student |   Teacher    | $AP^{bb}$ | $AP^{bb}_{50}$ | $AP^{bb}_{75}$ | $AP^{mk}$ | $AP^{mk}_{50}$ | $AP^{mk}_{75}$ |
> | ------ | :-----: | :----------: | --------- | -------------- | -------------- | --------- | -------------- | -------------- |
> | SEED   |  R-34   | R-152 (74.1) | 38.4      | 57.0           | 41.0           | 33.3      | 53.7           | 35.3           |
> | DisCo  |  R-34   | R-152 (74.1) | 39.4      | 58.7           | 42.7           | 34.4      | 55.4           | 36.7           |
> | BINGO  |  R-34   | R-152 (74.1) | **39.9**  | **59.4**       | **43.5**       | **35.7**  | **56.5**       | **38.2**       |
>
> We further evaluate the generalization of BINGO on CIFAR-10/100 classification tasks. For fair comparison, we use ResNet-18 distilled from the ResNet-50 (67.4% ImageNet accuracy) teacher. The results are shown as follows:
>
> | Method | Student |   Teacher   | CIFAR-10 Top-1 Acc | CIFAR-100 Top-1 Acc |
> | :----: | :-----: | :---------: | :----------------: | :-----------------: |
> |  SEED  |  R-18   | R-50 (67.4) |        82.3        |        56.8         |
> | DisCo  |  R-18   | R-50 (67.4) |        85.3        |        63.3         |
> | BINGO  |  R-18   | R-50 (67.4) |      **86.8**      |      **66.5**       |

---

> ### Author Response · Authors · 2021-11-21
> **Thanks and Response to Reviewer dQqV (2)**
>
> > Q4: Other comparison that is, in my opinion, not very conclusive is the semi supervised learning on imagenet. Other works (e.g. DisCo) show a graph of performance vs teacher. The largest teacher yields better performance than the one shown here, but I didn't see a clarification of which teacher is used.
>
> **A4**: Thank you for this reminder. We supplement the following experimental results to fairly compare with SEED and DisCo on semi-supervised evaluation. Our method consistently outperforms both SEED and DisCo with large margins under different student-teacher settings.
>
> | Method | Student |       Teacher        | 1% Labels | 10% Labels |
> | ------ | :-----: | :------------------: | :-------: | :--------: |
> | SEED   |  R-18   |     R-50 (67.4)      |   39.1    |    50.2    |
> | DisCo  |  R-18   |     R-50 (67.4)      |   39.2    |    50.1    |
> | BINGO  |  R-18   |     R-50 (67.4)      | **42.8**  |  **57.5**  |
> | SEED   |  R-18   |     R-152 (74.1)     |   44.3    |    54.8    |
> | DisCo  |  R-18   |     R-152 (74.1)     |   47.1    |    54.7    |
> | BINGO  |  R-18   |     R-152 (74.1)     | **50.3**  |  **61.2**  |
> | BINGO  |  R-18   | R-50$\times$2 (77.3) |   48.2    |    60.2    |
>
>
>
> > Q5: MobileNet family not considered.
>
> **A5**: In the original manuscript, we have evaluated EfficientNet-B0 (which is also in the MobileNet-family) distilled by ResNet-50$\times$2 and shown the result in Fig. 1 (a). BINGO achieves 69.3% top-1 accuracy, outperforming DisCo (69.1%) and SEED (67.6%) with the same teacher. In addition, we distill MobileNet-v3 for 200 epochs with the ResNet-50$\times$2 teacher. The results are summarized in the below table.
>
> | Method |     Student     |    Teacher    | Distill Epoch | Top-1 Acc |
> | :----: | :-------------: | :-----------: | :-----------: | :-------: |
> |  SEED  | EfficientNet-B0 | R-50$\times$2 |      800      |   67.6    |
> | DisCo  | EfficientNet-B0 | R-50$\times$2 |      200      |   69.1    |
> | BINGO  | EfficientNet-B0 | R-50$\times$2 |      200      | **69.3**  |
>
> | Method |   Student    |    Teacher    | Distill Epoch | Top-1 Acc |
> | :----: | :----------: | :-----------: | :-----------: | :-------: |
> |  SEED  | MobileNet-v3 | R-50$\times$2 |      800      | **68.2**  |
> | DisCo  | MobileNet-v3 | R-50$\times$2 |      200      |   58.9    |
> | BINGO  | MobileNet-v3 | R-50$\times$2 |      200      |   67.1    |
>
> BINGO outperforms DisCo on both EfficientNet-B0 and MobileNet-v3 architectures. In addition, when distilling EfficientNet-B0, BINGO achieves a better result (69.3% accuracy) with 200 distillation-epochs compared with SEED with 800 distillation-epochs. Although BINGO's result with MobileNet-v3 is slightly worse than SEED, we deduce that MobileNet-v3 is much smaller and harder to optimize, which needs more distillation-epochs but takes more computation cost. We would like to try with more epochs in the future.
>
> >  Q6: The related work focuses more on pretext tasks than on contrastive method. This could do with some rebalancing. I'd suggest citing the conference paper rather than the arxiv version.
>
> **A6**: Thanks for your advice, we will discuss about more related works about contrastive method and check all citations carefully in our revised version.

---

> ### Author Response · Authors · 2021-11-21
> **Thanks and Response to Reviewer dQqV (3)**
>
>
> > Q7: Why on Table 1, the first two lines of penultimate block have a different teacher performance than the 3rd line?
>
> **A7**: Both 67.4%- and 71.1%-accuracy checkpoints of the ResNet-50 teacher model can be directly downloaded from the [MoCo github repository](https://github.com/facebookresearch/moco). We simply choose a better-performed model as the teacher for distillation, and also choose ResNet-50$\times$2 with the same accuracy as the teacher for fair comparison in the original submission. We also supplement the results with the same ResNet-50 teacher with 67.4% accuracy and ResNet-152 teacher with 74.1% accuracy as follows and the advantage of our method is still evident.
>
> | Method |   Teacher   |   R-18   |   R-34   |
> | :----: | :---------: | :------: | :------: |
> |  SEED  | R-50 (67.4) |   57.6   |   58.5   |
> | DisCo  | R-50 (67.4) |   60.6   |   62.5   |
> | BINGO  | R-50 (67.4) | **61.4** | **63.5** |
> | BINGO  | R-50 (71.1) | **64.0** | **66.1** |
>
> | Method |   Teacher    |   R-18   |   R-34   |
> | :----: | :----------: | :------: | :------: |
> |  SEED  | R-152 (74.1) |   59.5   |   62.7   |
> | DisCo  | R-152 (74.1) |   65.5   |   68.1   |
> | BINGO  | R-152 (74.1) | **65.9** | **69.1** |
>
>
> > Q8: Table 7, 3rd line, looks like a very weird results. Are your sure of that one?
>
> **A8**: The setting of the 3rd row in Tab. 7 is that we replace the pretrained teacher model $f_\mathbf{T}$ of Eq.(8) and Eq.(9) with $f_{\mathbf{S}}$, which follows EMA-updating as in the standard MoCo setting. The pretrained teacher model is only used for generating data relation, i.e., selecting positive samples $\mathbf{x_p}$ for $\mathbf{x_a}$. This result is reasonable which exactly demonstrates the core idea of our paper that BINGO can achieve a promising performance only using teacher's relation knowledge, i.e. 62.2% accuracy in Tab. 7.
>
>
>
> [R1] Khosla, Prannay, et al. "Supervised Contrastive Learning." Advances in Neural Information Processing Systems 33 (2020).
>
> [R2] Fang, Zhiyuan, et al. "SEED: Self-supervised Distillation For Visual Representation." *ICLR*, 2020.
>
> [R3] Gao, Yuting, et al. "DisCo: Remedy Self-supervised Learning on Lightweight Models with Distilled Contrastive Learning." *arXiv:2104.09124* (2021).

---

> ### Author Response · Authors · 2021-11-26
> **Thanks and Looking Forward to Your Reply**
>
> Dear Reviewer,
>
> we would appreciate your review work which helps us a lot in improving our paper. We wonder if there is any additional question and would like to answer at any time. If our response has cleared your concerns, could you please consider raising the rating?
>
> Many thanks for your comments again.

---

> > ### Comment · Reviewer_dQqV · 2021-11-29
> > **Thanks for a thorough rebuttal**
> >
> > I have to say the number of experiments added is impressive. It is surprising that they were not included in the original manuscript. The object detection experiments are indeed compelling. The experiments with EfficientNet and MNV3 are appreciated too, but they highlight the limitations of the rebuttal phase as a period to run key experiments. I'm happy with the added evidence and I'll raise the score (I know the reviewing process is sometimes frustrating, but I'd like to hope this is not something you need to actively ask for)
> >
> > Just some extra comments:
> > A1: I understand how your method is different from the supervised contrastive learning, just thought it might be an interesting relation to consider.
> >
> > A2-A3: yes, I understood the rationale... but it is also relatively easy to make a credible argument for the contrary. The question is how things work out in practice, which is well answered with A3. I'm still not sure why this was not part of the main manuscript.
> >
> > A4: Very good results
> >
> > A5: Not so good results, but admittedly a bit on the inconclusive side given the different number of epochs. I don't mind, as long as it is shown in the paper. Limitations are ok, but they should be in the open so the reader understands.
> >
> > A6-A8 thanks for the clarifications (I think the "minor comments" part of my initial review got clumped together, hopefully that didn't cause confusion)

---

> > > ### Author Response · Authors · 2021-11-29
> > > **Thanks for Your Kind Response**
> > >
> > > Dear Reviewer,
> > >
> > > we sincerely appreciate your detailed comments. We would admit the rebuttal process is sometimes a little "frustrating", but it actually helps us improve our paper and makes us happy in turn. Some perspectives were not taken into account in our initial submission but indicated by several kind reviewers. As we read your item-by-item comments, we could feel our response actually clarified these questions. And your kind comments do make us feel warm in the heart.
> > >
> > > Certainly, we will carefully revise our paper as your suggestions in the final version.

---

### Official Review · Reviewer_5UEk · 2021-11-03

**Correctness:** 4
**Technical Novelty And Significance:** 3
**Empirical Novelty And Significance:** 4
**Recommendation:** 8
**Confidence:** 3

**Main Review:**

The BINGO method seems to be simple conceptually, yet yields good results on unsupervised pretraining on small network.

Positive points
+ well exposed, although some aspects would benefit from more detail, see below
+ interesting experiments, visualizations and ablation studies

A few points would strengthen the paper:
1) eq. (6) does not make use of data augmentation, yet its description mentions data augmentations, please clarify where it is used; are each instance different augmentations of an instance, for the intra as well as for the inter-sample loss?

2) eq. (8) make negative samples appear, but it is not clear how these samples are selected and they are not referred to in the text, please make clear for self-containment.

3) It would be interesting to highlight the accuracy of the Teacher model for reference in some of the tables.

4) p.7 "In our experiments, both the data relation and model parameters of teacher model are used to distill student model". I am not sure what this means. Does the "model parameters" refer to the Lintra part of the loss, and the "data relation" refers to the Linter part of the loss, and is this an ablation of the impact of the two components of the loss? Please clarify this experiment.

Minor:
a) Language is understandable, but could benefit from further proofreading, e.g. in the abstract: "targets at transferring" -> "aims at transferring"; or p.2 "This may be heuristic for" -> This might be a useful insight for... (for example); p.3: "rare of them pay attention" -> "few of them pay attention"; etc.

b) p.2: the explanation for inter-sample and intra-sample distillation components are inverted in the introduction.

c) Reference [Caron et al. 2020] was published in NeurIPS 2020 and there is no reason to use the arXiv version in the citation. Please double-check the other citations as well.

**Summary Of The Paper:**

The paper introduces a method called BINGO for unsupervised feature learning of small models that rely on the distillation from larger teacher networks, i.e. unsupervised distillation.
BINGO distills knowledge from a large model pretrained in an unsupervised manner to a small student network.
The main novelty is that the distillation from the teacher to the student leverages the relations learned by the teacher on the dataset, i.e. aims to transfer similarity clusters formed by K nearest neighbours.
The unsupervised pretraining of the student models is evaluated with linear or K-NN classification on ImageNet, in a limited labeled data setting, or by assessing generalizability to detection and instance segmentation. The method obtains competitive results w.r.t the state of the art with small models.
Ablation studies are done to investigate the impact of the different ingredients of the method.

**Summary Of The Review:**

The paper presents an interesting method and good supporting experiments. I would like to see the few points of clarification I raised above addressed in the rebuttal.

---

> ### Author Response · Authors · 2021-11-21
> **Thanks and Response to Reviewer 5UEk (1)**
>
> We sincerely thank you for your feedback and hope our following clarifications and responses could clear your concerns.
>
> > Q1: Eq. (6) does not make use of data augmentation, yet its description mentions data augmentations, please clarify where it is used; are each instance different augmentations of an instance, for the intra as well as for the inter-sample loss?
>
> **A1**: Sorry for this unclear clarification. As shown in Fig. 2, the anchor instance is pre-processed into two views and the positive instance is transformed into one view, both using the standard MoCo-v2-style augmentations before fed into the student and teacher network. We revise the Eq. (6) of initial submission as follows:
> $$
> \mathcal L = L(f_\mathbf{S}(\mathbf{t_1}(\mathbf{x_a})), f_\mathbf{T}(\mathbf{t_2}(\mathbf{x_a}))) + \mathop {\mathbb E}_{\mathbf{x_i} \sim \mathbf{\Omega_a} \setminus \mathbf{x_a}} (L(f_\mathbf{S}(\mathbf{t_3}(\mathbf{x_i})), f_\mathbf{T}(\mathbf{t}_2(\mathbf{x_a})))),
> $$
> where three independent data augmentation operators $\mathbf{t}_1, \mathbf{t}_2, \mathbf{t}_3$ are randomly sampled from the same family of MoCo-v2[R1] augmentations $\mathcal T$, which is also adopted in SEED[R2] and DisCo[R3].
>
>
> > Q2: Eq.(8) make negative samples appear, but it is not clear how these samples are selected and they are not referred to in the text, please make clear for self-containment.
>
> **A2**: We select negative samples in a memory bank, which is widely used in MoCo[R4] and many subsequent contrastive learning methods. The memory bank is a queue of data embeddings and the queue size is much larger than the typical mini-batch size. After each forward iteration, the embeddings in the queue are progressively replaced by the current output of the teacher network. In Eq. (8) and Eq. (9) of revised submission, we use embeddings in the above mentioned memory bank as negative samples. We will supplement this detail in the revised version.
>
> > Q3: It would be interesting to highlight the accuracy of the Teacher model in some of the tables.
>
> **A3**: Thank you for this kind reminder. In the revised paper, we will add the teacher accuracy in Tab. 1,4,5 in the revised submission for clearer presentation.

---

> > ### Comment · Reviewer_5UEk · 2021-12-01
> > **Thanks for the clarifications**
> >
> > I thank the authors for the clarifications they have provided and I find the approach and results compelling, and made stronger by all the additional experiments provided in the rebuttal (such as the fair comparison for the R-50 teacher). I am therefore keeping my initial positive rating.

---

> > > ### Author Response · Authors · 2021-12-01
> > > **Thanks for Your Response**
> > >
> > > We would appreciate your kind response. Many thanks for your efforts and recommendations which make our paper better.

---

> ### Author Response · Authors · 2021-11-21
> **Thanks and Response to Reviewer 5UEk (2)**
>
> > Q4: p.7 "In our experiments, both the data relation and model parameters of teacher model are used to distill student model". I am not sure what this means. Does the "model parameters" refer to the Lintra part of the loss, and the "data relation" refers to the Linter part of the loss, and is this an ablation of the impact of the two components of the loss? Please clarify this experiment.
>
> **A4**: Sorry for this ambiguousness. **model parameters** represent whether to load the teacher model parameters into the "Teacher" part in Fig. 2. **data relation** refers to the $\mathcal L_{inter}$ loss. Besides, $\mathcal{L}_{intra}$ is a basic loss in contrastive-learning paradigm on two views of one instance.
>
> We list the detailed loss function with different **model parameters** and **data relation** settings for clear demonstration as follows.
>
> Eq. R1: w/ **model parameters** & w/ **data relation**
>
>    This setting is the standard experimental setting in this paper and the loss function is as
> $$
> \mathcal L = - \log \frac{ \exp (f_\mathbf{S}(\mathbf{t_1}(\mathbf{x_a})) \cdot f_\mathbf{T}(\mathbf{t_2}(\mathbf{x_a}))  /\tau ) } {\sum_{i=0}^{N} \exp (f_\mathbf{S}(\mathbf{t_1}(\mathbf{x_a})) \cdot {\mathbf{k_i^-} }/\tau )} - \log \frac{ \exp (f_\mathbf{S}(\mathbf{t_3}(\mathbf{x_p})) \cdot f_\mathbf{T}(\mathbf{t_2}(\mathbf{x_a}))  /\tau ) } {\sum_{i=0}^{N} \exp (f_\mathbf{S}(\mathbf{t_3}(\mathbf{x_p})) \cdot {\mathbf{k_i^-} }/\tau )}
> $$
>    where $f_\mathbf{T}$ denotes the pretrained teacher network (R-50/R-152/R-50x2).
>
> Eq. R2: w/ **model parameters** & w/o **data relation**
>
>    This setting is the conventional distillation setting, i.e., only $\mathcal{L}_{intra}$ loss is used for distillation. The total loss function is as
>
> $$
> \mathcal L = - \log \frac{ \exp (f_\mathbf{S}(\mathbf{t_1}(\mathbf{x_a})) \cdot f_\mathbf{T}(\mathbf{t_2}(\mathbf{x_a}))  /\tau ) } {\sum_{i=0}^{N} \exp (f_\mathbf{S}(\mathbf{t_1}(\mathbf{x_a})) \cdot {\mathbf{k_i^-} }/\tau )}.
> $$
>
> Eq. R3: w/o **model parameters** & w/ **data relation**
>
>    In this setting, we replace the pretrained teacher network $f_\mathbf{T}$ in the loss function of setting-1 with $f_{\mathbf{S}}$, which follows EMA-updating as in the standard MoCo setting. The total loss is as
> $$
> \mathcal L = - \log \frac{ \exp (f_\mathbf{S}(\mathbf{t_1}(\mathbf{x_a})) \cdot f_{\mathbf{S}}(\mathbf{t_2}(\mathbf{x_a}))  /\tau ) } {\sum_{i=0}^{N} \exp (f_\mathbf{S}(\mathbf{t_1}(\mathbf{x_a})) \cdot {\mathbf{k_i^-} }/\tau )} - \log \frac{ \exp (f_\mathbf{S}(\mathbf{t_3}(\mathbf{x_p})) \cdot f_{\mathbf{S}}(\mathbf{t_2}(\mathbf{x_a}))  /\tau ) } {\sum_{i=0}^{N} \exp (f_\mathbf{S}(\mathbf{t_3}(\mathbf{x_p})) \cdot {\mathbf{k_i^-} }/\tau )}
> $$
>    This first term is the conventional contrastive learning loss and the second term is a variant of $\mathcal{L}_{inter}$ loss, the $\mathbf{x}_p$ is the positive samples obtained by the pretrained teacher network.
>
> We have shown ablation results on the impact from the two components, i.e., "model parameters" and "data relation", in Tab. 7. The 3rd row with 62.2% accuracy corresponds to the result of only using data relation (Eq. R3 in the above formulation). The last row with 64.0% accuracy corresponds to the result of using both teacher model parameters and data relation (Eq. R1). We also supplement an additional ablation study with only teacher parameters (Eq. R2). The results are summarized in the below table:
>
> | Teacher Model Parameters | Data Relation | Accuracy(%) |
> | :----------------------: | :-----------: | :---------: |
> |       $\checkmark$       |               |    62.0     |
> |                          | $\checkmark$  |    62.2     |
> |       $\checkmark$       | $\checkmark$  |  **64.0**   |
>
>
> We have added the result and revised the representations of Tab. 7 in the revised submission.
>
> > Q5: Language and reference.
>
> **A5**: Thanks for your kind suggestions and we will revise our paper carefully and check all citations' correctness.
>
>
>
> [R1] Chen, Xinlei, et al. "Improved baselines with momentum contrastive learning." *arXiv:2003.04297* (2020).
>
> [R2] Fang, Zhiyuan, et al. "SEED: Self-supervised Distillation For Visual Representation." *ICLR*, 2020.
>
> [R3] Gao, Yuting, et al. "DisCo: Remedy Self-supervised Learning on Lightweight Models with Distilled Contrastive Learning." *arXiv:2104.09124* (2021).
>
> [R4] He, Kaiming, et al. "Momentum contrast for unsupervised visual representation learning." *CVPR*, 2020.

---

### Author Response · Authors · 2021-11-22
**To All Chairs and Reviewers**

Dear Chairs and Reviewers,

We sincerely thank all your efforts on review works and all the valuable comments. We have analyzed these review comments sentence by sentence carefully and revised our paper following the suggestions. All the revised contents in the paper have been colored in blue and the main changes are summarized as follows.

1. We check all citations' correctness and add some related works.
2. MobileNet-v3 results are added in Fig. 1 (a).
3. Eq. (6) is revised to include specific data augmentation transformations.
4. How negative samples are selected is clarified in Sec. 3.2.
5. We add ImageNet classification results of student models distilled from ResNet-50 and ResNet-152 teachers with the same accuracy as previous methods for fair comparisons in Tab. 1.
6. Tab. 3 is added to make a fair comparison with SEED and DisCo on downstream tasks of object detection and instance segmentation.
7. We add ImageNet semi-supervised classification results of ResNet-18 distilled from ResNet-50 and ResNet-152 teachers with the same accuracy as previous methods for fair comparisons in Tab. 4.
8. Evaluation of downstream tasks on CIFAR-10 and CIFAR-100 classification is supplemented in Tab. 5.
9. One additional ablation study of only using teacher parameters is provided in Tab. 7.
10. We add analysis of computational complexity in Sec. 4 and provide comparisons with SEED and DisCo under the same computation cost in Tab. 9.

We hope our responses could clear concerns indicated in the review with respect and are happy to answer any additional questions at any time.

---

### Decision · Program_Chairs · 2022-01-20

**Decision:**

Accept (Poster)

**Comment:**

Four experts reviewed the paper. All but Reviewer HSTU recommended acceptance. The authors clearly did a great job with the rebuttal, which convinced two reviewers to raise their scores above the acceptance threshold. Notably, the reviewers found the newly added experiments impressively strong. The rebuttal also addressed some clarification questions. Based on the reviewers' feedback, the decision is to recommend the paper for acceptance. As mentioned by the reviewers, some experiments and discussions in the rebuttal should be included in the paper. The authors are encouraged to make the necessary changes to the best of their ability. We congratulate the authors on the acceptance of their paper!